# TGRS: Teacher-Guided Rank-Sensitive Quantization for Large Language Models

## Abstract

Compression techniques such as quantization and low-rank approximation have enabled large language models (LLMs) to run on resource-constrained hardware, but they often fall short in capturing the heterogeneous sensitivity of model components. In this paper, we propose **Teacher-Guided Rank Sensitivity (TGRS)**, a novel LLM compression framework that uses a data-informed, direction-level sensitivity profiler to directly quantify parameter importance with respect to prediction accuracy and representational capacity. By projecting the importance signals onto the singular directions of weight matrices, TGRS establishes a principled scoring mechanism that drives a global, budget-aware allocation strategy. The allocator dynamically determines where low-rank corrections are most effective, preserving expressivity in sensitive regions while aggressively compressing redundant ones. TGRS requires no retraining or custom kernels, yet consistently outperforms prior methods. Most notably, on LLaMA-3.1-8B at an aggressive 3.6-bit budget, TGRS achieves 4.4× compression with minimal perplexity degradation from 6.63 to 6.78 (+0.15). By reducing memory requirements from 16GB to 3.9GB, models can be easily deployed on edge devices like Jetson Orin Nano (8GB memory).

## 1 Introduction

Large language models (LLMs) such as GPT-4 (OpenAI, 2023), Gemini (Anil et al., 2023), and LLaMA-2/3 (Touvron et al., 2023a) demonstrate competitive performance on language and multi-modal tasks, but inference typically assumes high memory bandwidth, ample storage, and optimized kernels. By contrast, edge and commodity devices operate under strict VRAM and scheduling constraints, so post-training compression is commonly used for deployment (Jaiswal et al., 2024).

To this end, the classical axes of compression, pruning, quantization, and low-rank factorization for machine learning models have all progressed substantially. Pruning can remove redundancy (Han et al., 2016; Frankle & Carbin, 2019; Frantar & Alistarh, 2023); quantization enables low-precision inference with calibration and second-order refinements (Frantar et al., 2022; Dettmers et al., 2022b; Lin et al., 2024; Esser et al., 2020; Hubara et al., 2017); and low-rank methods exploit matrix structure (Hu et al., 2022; Denton et al., 2014; Trockman & Kolter, 2021; Aghajanyan et al., 2021). At the billion-parameter scale, however, layers and projections exhibit *heterogeneous sensitivity* to compression: some regions remain quality at very low precision while others importance gain from targeted protection (Frantar & Alistarh, 2023; Dettmers et al., 2022a; Xiao et al., 2023). Recent works on machine learning systems further underscore memory and scheduling as primary bottlenecks for serving (Dao et al., 2022), and edge-oriented models (Sun et al., 2020; Tambe et al., 2021), which also highlight the need to consider both quality and deployability. Motivated by the goal of preserving text quality (lower perplexity) under stringent bit budgets, we adopt a global allocation strategy driven by a multi-faceted importance score that synthesizes insights from the functional, optimization, and structural domains, thereby enabling selective preservation of the most critical directions under a quantized backbone.

In this paper, we introduce **Teacher-Guided Rank Sensitivity (TGRS)**, a post-training framework that treats compression as a global allocation problem under an explicit storage budget. We adopt a *planning* viewpoint: instead of asking how to compress each layer in isolation, we allocate a *single global budget* to the directions that matter most for accuracy under post-training perturbations. The overview of TGRS is shown in Figure 1. Concretely, we map direction-level scores in the SVD basis

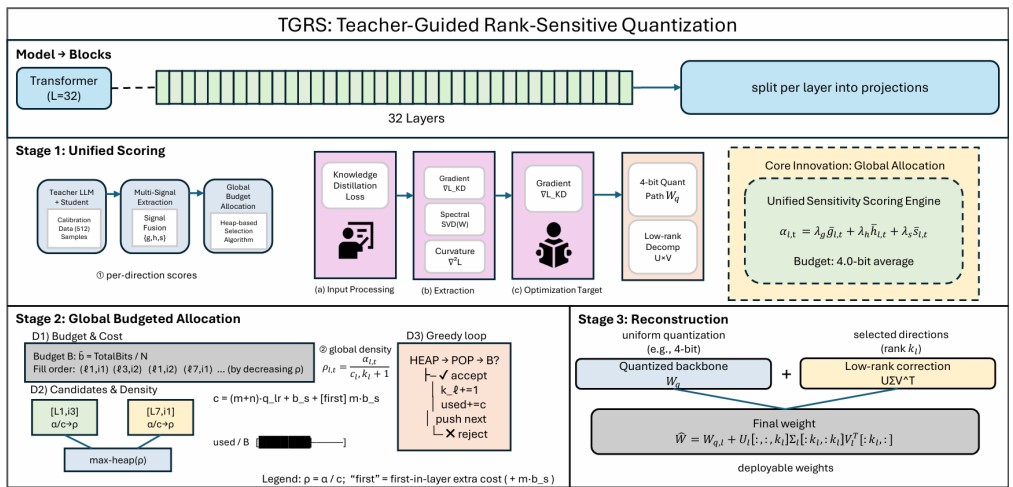

Figure 1: Overview of the TGRS framework. (A) Split transformer layers into projection matrices. (B) Score singular directions using gradients, curvature, and spectral energy, then normalize and combine. (C) Allocate a global bit budget by selecting directions with the highest importance-per-bit. (D) Reconstruct the model by adding selective low-rank corrections to a quantized backbone.

into a unified sensitivity score and formulate compression as a global allocation problem: the budget is invested in the most high-impact directions, while the remaining components are quantized more aggressively. Our main technical contributions are summarized as follows:

- *Quantization under a storage budget.* TGRS achieves low perplexity under tight bit budgets while maintaining a uniformly quantized backbone.

- *Direction-level sensitivity scoring.* We identify compression-sensitive directions in weight matrices before compression by projecting observable signals onto singular directions, yielding a ranked list of selection.

- *Global budgeted allocator.* We formulate rank selection as an importance-per-bit allocation under a budget that includes both quantized weights and metadata (uniformly quantized backbone weights and selective low-rank residuals).

- *Evaluation and deployment.* We evaluate the performance comprehensively on various benchmarks. For LLaMA-3.1-8B on Jetson Orin Nano (8 GB), TGRS is selectively placed low-rank corrections over a quantized backbone to reach 0.78–0.92 s first-token latency, 1.1–1.3 tok/s steady-state, with 3.5–3.8 GB resident memory (Table 5).

## 2 RELATED WORK

**Post-training quantization (PTQ): PTQ and calibration strategies.**  Transformer PTQ studied from calibration heuristics to structure-aware, low-bit regimes. Representative advances include INT8 with outlier handling (Dettmers et al., 2022a), second-order layerwise optimization (Frantar et al., 2022), activation-aware channel protection (Lin et al., 2024), activation smoothing for joint W + A quantization (Xiao et al., 2023), and efficient group-wise pipelines (Yao et al., 2022). Recent works have explored sub-4-bit operation via codebooks, orthogonalization, or lattice formulations (Tseng et al., 2024; Egiazarian et al., 2024; Eliseev & Mazur, 2023; Ashkboos et al., 2024; Zhang et al., 2024).

**Hybrid quantization with structured residuals: Hybrid quantization with structured residuals.** To tighten quality–compression performance under different budgets, hybrid schemes maintain a quantized backbone while enabling small, structured corrections where beneficial. Examples include spectral-structure-aware low-rank corrections (Saha et al., 2024), co-design of quantization and adapter initialization (Li et al., 2024; Lawton et al., 2024), serving-oriented low-bit designs (Zhao et al., 2024), and elastic precision scheduling (Lee et al., 2023; Kwon et al., 2022). Collectively,

these works motivate mechanisms that *allocate capacity where it matters*. They share the principle of retaining a quantized backbone while adding small, structured corrections in selected locations, differing mainly in whether corrections address quantization residuals, the granularity of selection (layer, block, or direction), and the extent to which compatibility with standard kernels is preserved.

**Direction-aware protection and global allocation.**    A growing body of evidence indicates that compression sensitivity is *anisotropic within layers*. Hessian-guided sparsification (Frantar & Alistarh, 2023), magnitude/activation proxies (Sun et al., 2024), and block-wise second-order reconstruction (Li et al., 2021) all reveal direction-dependent sensitivity. At a coarser granularity, mixed-precision schedulers formalize bitwidth assignment as a budgeted optimization so that precision is redistributed toward components with higher predicted importance gain over an additional bit (Lee et al., 2023). Hybrid schemes further combine a low-bit backbone with structured residuals, using spectral cues (e.g., energy in leading singular directions) to decide where limited corrective capacity is applied (Saha et al., 2024). These methods extract *direction-aware* sensitivity signals (Hessian, activations, gradients, or spectral energy) and conduct *global allocation* of scarce precision or rank under an explicit budget.

## 3 Method

In this work, we target a training-free pipeline that exports a single model directly compatible with standard inference stacks. Our approach builds on direction-level profiling and global allocation applied post-training, with an explicit bit ledger to ensure comparability under a fixed budget. We compute gradients and an Exponential Moving Average curvature proxy only for scoring; model weights remain frozen throughout, with no weight updates or fine-tuning. Concretely, we cast post-training compression as a singular-direction global allocation problem under a storage budget: (i) decompose each block's quantization residual into singular directions as candidates, (ii) score each direction using three concrete and observable signals—teacher-guided gradients, a curvature proxy, and spectral energy, and (iii) run a global greedy allocator that spends a single budget on the most valuable directions. The output is a uniformly-quantized backbone corrected by sparse low-rank updates, directly compatible with standard inference stacks.

### 3.1 Quantization and Residual Formation

Let the set of linear projection matrices in a transformer be indexed by $l \in \{1, \dots, L\}$. Each layer is represented by a weight matrix $W_l \in \mathbb{R}^{m_l \times n_l}$. Our compression process begins with symmetric per-row quantization to a bit-width of $b_l$. This involves three steps to obtain the quantized backbone $W_{q,l}$ and its corresponding residual $R_l$. First, for each row $i$, we compute a quantization scale $s_{l,i}$ that maps the row's dynamic range to the maximum representable integer, $q_{\max} = 2^{b_l - 1} - 1$:

$$s_{l,i} \;=\; \frac{\max(\max_j |w_{l,i,j}|, \; \varepsilon)}{q_{\max}}, \tag{1}$$

where $\varepsilon$ is a small constant (e.g., $10^{-8}$) to avoid division by zero when the entire row is zero. where $w_{l,i,j}$ is the entry in the $i$-th row and $j$-th column of $W_l$. Second, using this scale, the floating-point weights are converted to an integer matrix, which we denote as $W_l^{\text{int}}$. Each element of this matrix, $w_{l,i,j}^{\text{int}}$, is computed via a rounding and clipping function:

$$w_{l,i,j}^{\text{int}} \;=\; \text{clip}\left(\left\lfloor \frac{w_{l,i,j}}{s_{l,i}} \right\rceil, \; -q_{\max}, \; q_{\max}\right). \tag{2}$$

Here, $\lfloor \cdot \rceil$ denotes rounding to the nearest integer, and $\text{clip}(x, a, b)$ confines $x$ to the range $[a, b]$. Then, the dequantized weight matrix, $W_{q,l} = S_l W_l^{\text{int}}$, which serves as the floating-point approximation of the original weights, is then recovered by applying the scales back to the integer matrix. Letting $S_l = \text{diag}(s_l)$ be the diagonal matrix formed from the row scales, this operation is defined as:

$$R_l \;=\; W_l - W_{q,l}. \tag{3}$$

Our goal is to approximate this residual with a low-rank matrix. To do this, we first analyze its structure using SVD, which factorizes $R_l$ as:

$$R_l \;=\; U_l \Sigma_l V_l^\top \;=\; \sum_{t=1}^{r_l} \sigma_{l,t} \, u_{l,t} v_{l,t}^\top, \tag{4}$$

Here, the columns of $U_l \in \mathbb{R}^{m_l \times r_l}$ and $V_l \in \mathbb{R}^{n_l \times r_l}$ are the left and right singular vectors, respectively, and $\Sigma_l$ is a diagonal matrix containing the singular values $\sigma_{l,t}$ in descending order, where the matrix rank is $r_l = \min(m_l, n_l)$. The goal of our method is to select an integer rank $k_l \in \{0, 1, \ldots, r_l\}$ for each layer, which specifies how many of these singular directions to use for the low-rank correction. The process for determining the optimal rank allocation vector $K = (k_1, \ldots, k_L)$ is the central topic of the following sections. To ensure numerical stability for layers assigned a 2-bit precision ($b_l = 2$), we apply a randomized block-orthogonal preconditioning to the original weight matrix $W_l$.

### 3.2 Stage 1: A Unified Importance Score for Global Allocation

A key property of LLMs is the heterogeneous sensitivity of their parameters: some are highly robust to compression, while others are extremely fragile. Thus, parameter importance is inherently multi-faceted rather than monolithic. For instance, a parameter may be crucial for influencing the final output (functional importance) yet appear less significant when judged solely by its raw magnitude (structural importance).

To capture this complexity, TGRS introduces a novel scoring function that synthesizes three complementary analytical viewpoints to holistically assess each singular direction. Our method uses two distinct Singular Value Decompositions (SVDs): one for reconstruction and one for scoring. Note that both SVDs are computed only once per layer and serve distinct purposes: residual SVD for reconstruction, scoring-basis SVD for projecting signals. To establish an orthogonal basis for scoring, we perform an SVD on a potentially preconditioned version of the original weight matrix, which we denote with a tilde ($\sim$):

$$\widetilde{W}_l = \widetilde{U}_l \widetilde{\Sigma}_l \widetilde{V}_l^\top. \tag{5}$$

This is distinct from the SVD of the residual matrix, $R_l$ (defined in Sec. 3.1), which is used for the final model reconstruction. The columns of $\widetilde{U}_l$ and $\widetilde{V}_l$ from this scoring-basis SVD serve as the basis vectors onto which we project our signals.

**Functional Criticality (via Teacher Gradients).** The "Teacher" in Teacher-Guided Rank Sensitivity (TGRS) is the cornerstone of our functional criticality score, providing a ground-truth signal for how the compressed model should behave. For our post-training framework, the teacher is simply the **original, full-precision model itself**. It acts as a perfect oracle, representing the ideal functional behavior we aim to preserve through compression.

We use this teacher to guide the process by measuring how perturbations in the compressed "student" model affect its output relative to the teacher. This is implemented using a standard knowledge distillation (KD) loss (Hinton et al., 2015), where on a small calibration set, $\mathcal{D}_{\text{calib}}$, we minimize the Kullback–Leibler (KL) divergence between the teacher's ($p_{\text{teacher}}$) and student's ($p_{\text{student}}$) output distributions:

$$\mathcal{L}_{\text{KD}} = \sum_{x \in \mathcal{D}_{\text{calib}}} \text{KL}(p_{\text{teacher}}(x) \,\|\, p_{\text{student}}(x)). \tag{6}$$

The functional importance score, $g_{l,t}$, is then the magnitude of the gradient of this loss projected onto a specific singular direction. Here, $t \in \{1, \ldots, r_l\}$ is the index of the singular direction, and $\widetilde{u}_{l,t}$ and $\widetilde{v}_{l,t}$ are the corresponding left and right singular vectors from the scoring-basis SVD:

$$g_{l,t} = \left| \widetilde{u}_{l,t}^\top \frac{\partial \mathcal{L}_{\text{KD}}}{\partial W_l} \widetilde{v}_{l,t} \right|. \tag{7}$$

We project onto the scoring-basis singular vectors $\widetilde{u}l, t, \widetilde{v}l, t$ (Eq. 5) rather than residual SVD vectors, so that all signals are evaluated in a consistent orthogonal basis

**Optimization Sensitivity (via Curvature Proxy).** To assess a direction's optimization sensitivity, we analyze the local geometry of the loss landscape. It is well-established that high-curvature regions are notoriously intolerant to perturbation (LeCun et al., 1989), as small parameter changes can induce large increases in the loss. We estimate this curvature using an efficient proxy for the Hessian diagonal, calculated via a coefficient $\beta$, Here $\beta \in [0, 1)$ is the Exponential Moving Average momentum (we use $\beta=0.9$ in all experiments), and $m_l$ denotes the number of rows of $W_l$:

$$H_{l,i,j}^{\text{diag}} \approx \beta H_{l,i,j}^{\text{diag}} + (1-\beta) \left( \frac{\partial \mathcal{L}}{\partial W_{l,i,j}} \right)^2. \tag{8}$$

To maintain efficiency, we compute a row-wise proxy, $\bar{h}_{l,i} = \sum_j H_{l,i,j}^{\text{diag}}$. The final optimization sensitivity score for the $t$-th singular direction, $h_{l,t}$, is the projection of this row-wise curvature proxy onto the left singular vectors over the $m_l$ rows of the matrix:

$$h_{l,t} = \sum_{i=1}^{m_l} \bar{h}_{l,i} \, (\widetilde{u}_{l,t,i})^2. \tag{9}$$

**Structural Importance (via Spectral Energy).** To measure a direction's structural importance, we examine its spectral properties. A direction's singular value corresponds to the variance it captures in the linear transformation, a core principle of low-rank approximation (Denton et al., 2014). Its share of the total spectral energy, $s_{l,t}$, serves as a direct measure of its structural contribution:

$$s_{l,t} = \frac{\widetilde{\sigma}_{l,t}^2}{\sum_{j=1}^{r_l} \widetilde{\sigma}_{l,j}^2}. \tag{10}$$

where $\widetilde{\sigma}_{l,t}$ is the $t$-th singular value of $\widetilde{W}_l$.

**Synthesis into a Unified Score.** The three raw signals are first scaled to a common $[0, 1]$ range via min–max normalization, denoted as $\{\bar{g}_{l,t}, \bar{h}_{l,t}, \bar{s}_{l,t}\}$. They are then unified into a single score, $\alpha_{l,t}$, using equal non-negative aggregation coefficients $(\lambda_g, \lambda_h, \lambda_s) = (0.5, 0.1, 0.4)$.

$$\alpha_{l,t} = \lambda_g \bar{g}_{l,t} + \lambda_h \bar{h}_{l,t} + \lambda_s \bar{s}_{l,t}. \tag{11}$$

This unified score provides a stronger and comprehensive importance landscape than any single metric alone. By jointly incorporating functional, optimization, and structural signals, it enables the global allocator to make more informed decisions about where fidelity should be preserved.

### 3.3 Stage 2: Global Budgeted Allocation

#### 3.3.1 Budget Formulation

To enable fair comparison under a unified storage constraint, we formulate optimization with respect to an *budget*, which explicitly accounts for both quantized integers and all associated metadata (e.g., per-row scales, low-rank factors). This formulation ensures that methods with heterogeneous metadata costs are evaluated on a rigorously comparable basis.

The total storage cost, $C(K)$, is the sum of a fixed backbone cost and a variable low-rank correction cost. We denote by $b_l$ the bit-width of the quantized backbone weights, by $b_s$ the bit-width used to store per-row scales, and by $b_r$ the bit-width for the low-rank factor matrices; unless otherwise specified, we set $b_l$=4, $b_r$=4, and $b_s$=16 in all experiments. The backbone cost for all layers is $\sum_{l=1}^{L} (b_l m_l n_l + b_s m_l)$, while the low-rank cost depends on the allocated ranks $\{k_l\}$ and is built upon a precise marginal cost. Specifically, the additional storage cost of adding one singular direction (i.e., increasing the correction rank by one) to layer $l$ depends on whether this is the first assigned direction for that layer ($k_l : 0 \rightarrow 1$). For the first assigned direction, it incurs a one-time overhead to store per-row scales for one factor matrix; subsequent additions pay only the per-direction cost. We therefore define the marginal cost of increasing the rank from $k$ to $k+1$ as:

$$c_{l,k+1} = (m_l + n_l)b_r + 2b_s + \mathbb{I}[k = 0] \cdot m_l b_s. \tag{12}$$

Here, $(m_l + n_l)b_r + 2b_s$ is the base cost for adding one direction, and the term $\mathbb{I}[k{=}0] \, m_l b_s$ accounts for a one-time per-row scale cost only for the first addition (i.e., when moving from $k = 0$ to $k = 1$), where $\mathbb{I}[k{=}0] = 1$ if $k = 0$ and 0 otherwise.

The total cost function $C(K)$ is then the sum of the backbone cost and the cumulative marginal costs of all allocated ranks:

$$C(K) = \sum_{l=1}^{L} (b_l m_l n_l + b_s m_l) + \sum_{l=1}^{L} \sum_{t=1}^{k_l} c_{l,t}. \tag{13}$$

Let $N = \sum_{l=1}^{L} m_l n_l$ denote the total number of scalar parameters, where $m_l$ and $n_l$ are the row and column dimensions of weight matrix $W_l$. the average bits, defined as the total stored bits divided by $N$ for a given allocation $K$, denoted $\bar{b}(K)$, is:

$$\bar{b}(K) = \frac{C(K)}{N}. \tag{14}$$

Given a target budget $B$, we consider rank selection as maximizing total unified importance subject to this budget:

$$\underset{K \in \mathbb{Z}_{\geq 0}^L}{\text{maximize}} \quad \sum_{l=1}^{L} \sum_{t=1}^{k_l} \alpha_{l,t} \quad \text{s. t.} \quad \overline{b}(K) \leq B. \tag{15}$$

### 3.3.2 Global Budgeted Allocation

We solve the constrained optimization in Eq. 15 using a greedy knapsack-style procedure, where the algorithm iteratively selects singular directions according to their importance gain over an additional bit, ensuring that the bit budget is allocated to the most impaction candidates across layers. The importance gain of selecting the $t$-th singular direction of layer $l$ is its unified importance score, $\alpha_{l,t}$, while the marginal cost $c_{l,k_l+1}$ is the number of additional bits required to increase the rank of layer $l$ from its current allocation $k_l$ to $k_l + 1$, derived from Eq. 12. The central metric for our allocator is the importance-per-bit (utility per bit) score $\rho_{l,t}$: we use $\rho_{l,t}$ to order candidates in a global max-heap (higher is better). At each step, we pop the highest-$\rho$ candidate; if its marginal cost fits the remaining budget, we accept it (set $k_l = k_l + 1$), deduct the cost, and push that layer's next candidate with its updated $\rho$. This process repeats until the budget is exhausted or no candidate fits.

$$\rho_{l,t} = \frac{\alpha_{l,t}}{c_{l,k_l+1}}. \tag{16}$$

The allocation proceeds by maintaining a max-heap of rank-increase candidates—one per layer—prioritized by their density $\rho$. At each step, the highest-density candidate is selected; if it fits within the remaining budget, it is accepted, and the next candidate from the same layer is pushed into the heap. This process repeats until the budget is exhausted or no candidates remain.

### 3.4 Stage 3: Model Reconstruction

The final stage reconstructs the compressed model weights, $\widehat{W}_l$. This is done by adding a rank-$k_l$ approximation of the residual, $R_{k_l}$, back to the quantized backbone, $W_{q,l}$. $k_l$ is the per-layer rank selected by the global allocator in Stage 2 (Sec. 3.3.2) denote this allocation by $K = (k_1, \ldots, k_L)$ as the solution to Eq. 15. The rank-$k_l$ approximation is constructed using the top $k_l$ singular directions from the residual's SVD (the reconstruction basis):

$$\widehat{W}_l = W_{q,l} + R_{k_l} = W_{q,l} + U_l[:, : k_l] \Sigma_l[: k_l, : k_l] V_l^\top[: k_l, :]. \tag{17}$$

The resulting model is the collection of reconstructed layer weights $\{\widehat{W}_l\}_{l=1}^{L}$, which is then ready for deployment. The full TGRS pipeline is outlined in Algorithm 1.

## 4 Experiments

We evaluate TGRS to validate its effectiveness and practicality. We report (i) main results at matched budgets versus state-of-the-art baselines, (ii) ablation studies of key design choices, (iii) flexibility under mixed-precision schedules, and (iv) deployability on resource-constrained hardware.

### 4.1 Experimental Setup

**Models and Datasets.** Our primary evaluations are conducted on the LLaMA model family, specifically LLaMA-2-7B and LLaMA-3.1-8B. To demonstrate the generalizability of our method across different architectural families, we also conduct experiments on Mistral-7B. Perplexity (PPL) is the primary metric for LLM generation text quality, reported on the WikiText-2 and C4 benchmarks.

**Evaluation Metrics.** We evaluate performance using several standard metrics. We report the average bits-per-parameter ('Bits') following our comprehensive cost formulation in Eq. 14, and the corresponding Compression Ratio ('CR') relative to FP16 ($16/\overline{b}$). For runtime performance, we measure steady-state throughput ('Tok/s'), first-token latency ('Lat'), and resident VRAM.

---

**Algorithm 1** The TGRS Algorithm

---

**Require:** Weights $\{W_l\}_{l=1}^L$, Budget $B$, Signals $\{G_l, H_l^{\text{diag}}\}$, Hyperparameter.

**Ensure:** Reconstructed Weights $\{\widehat{W}_l\}_{l=1}^L$, Final Rank Vector $K$.

1: **//Stage 1: Unified Scoring**
2: **for** $l = 1 \ldots L$ **do**
3:     $\widetilde{U}_l, \widetilde{\Sigma}_l, \widetilde{V}_l \leftarrow \text{SVD}(\text{Precondition}(W_l))$             ▷ Get scoring basis from original weights
4:     Compute importance scores $\{\alpha_{l,t}\}$ using eq. 7-eq. 11.
5: **end for**
6: **//Stage 2: Global Budgeted Allocation**
7: $K \leftarrow (0, \ldots, 0);$      $C_{\text{used}} \leftarrow \sum_{l=1}^L (b_l m_l n_l + b_s m_l);$      $B_{\text{rem}} \leftarrow (N \cdot B) - C_{\text{used}}$
8: Initialize max-heap $\mathcal{H}$ with candidates $\{(\rho_{l,1}, l, t = 1)\}_{l=1}^L$ using density from eq. 16.
9: **while** $\mathcal{H} \neq \emptyset$ **and** $B_{\text{rem}} > 0$ **do**
10:     $(\rho, l^*, t^*) \leftarrow \mathcal{H}.\text{pop}()$                        ▷ Get candidate with highest density
11:     $c_{\text{marginal}} \leftarrow (m_{l^*} + n_{l^*})b_r + 2b_s + \mathbb{I}[k_{l^*} = 0] \cdot m_{l^*} b_s$
12:     **if** $c_{\text{marginal}} \leq B_{\text{rem}}$ **then**
13:         $k_{l^*} \leftarrow k_{l^*} + 1;$      $B_{\text{rem}} \leftarrow B_{\text{rem}} - c_{\text{marginal}}$
14:         **if** $k_{l^*} < r_{l^*}$ **then**
15:             $c_{\text{next}} \leftarrow (m_{l^*} + n_{l^*})b_r + 2b_s$
16:             $\mathcal{H}.\text{push}((\alpha_{l^*,k_{l^*}+1}/c_{\text{next}}, l^*, k_{l^*} + 1))$
17:         **end if**
18:     **end if**
19: **end while**
20: **//Stage 3: Reconstruction**
21: **for** $l = 1 \ldots L$ **do**
22:     $W_{q,l} \leftarrow \text{Dequantize}(\text{Quantize}(W_l));$      $R_l \leftarrow W_l - W_{q,l}$ ▷ Compute backbone and residual
23:     $U_l, \Sigma_l, V_l \leftarrow \text{SVD}(R_l)$                   ▷ Decompose residual for correction basis
24:     $\widehat{W}_l \leftarrow W_{q,l} + U_l[:, : k_l]\Sigma_l[: k_l, : k_l]V_l^\top[: k_l, :]$        ▷ Add rank-$k_l$ correction
25: **end for**
26: **return** $\{\widehat{W}_l\}, K$

---

**Baselines.** We compare to RTN (uniform round-to-nearest), GPTQ (Frantar et al., 2022), AWQ (Lin et al., 2024), CALDERA (Saha et al., 2024), LoftQ (Li et al., 2024), Atom (Zhao et al., 2024), and FlexQuant (Liu et al., 2025) under their recommended settings. The data-dependent signals for TGRS are derived using a small calibration set (64 sequences from WikiText-2 and 1000 from C4). The teacher-guided gradient signal is pre-computed on a diverse mix of instruction-tuning corpora, including Alpaca (Taori et al., 2023), Dolly (Conover et al., 2023), and OpenAssistant (Köpf et al., 2023)), to ensure general-purpose importance.

**TGRS Configurations.** We evaluate two configurations: (i) TGRS (Q4+LR), our main setting at the 4-bit operating point with a uniform 4-bit backbone and globally allocated low-rank corrections; and (ii) TGRS (Q-Mix), a high-compression variant that uses a mixed-precision backbone (e.g., 2–3 bits for less sensitive MLP projections) to reach tighter budgets (e.g., 3.6 bits).

### 4.2 Main Results

At 4-bit average budgets, *TGRS* consistently attains lower perplexity than uniform quantization (RTN) and remains competitive with or superior to other strong post-training baselines. We highlight two key configurations from our results in Table 1. Our primary configuration, TGRS (Q4+LR), is designed for a direct and fair comparison at the standard 4-bit operating point. To verify that TGRS's quality improvements do not increase runtime cost, we compare TGRS (Q4+LR) against the RTN Q4 baseline under identical kernels and settings. Table 2 shows matched throughput and VRAM, and latency that is comparable or lower. On LLaMA-2-7B, for instance, it successfully recovers over 50% of the perplexity gap between the uncompressed FP16 model and the naive round-to-nearest (RTN) baseline. On the other hand, the TGRS Q-Mix variant demonstrates the flexibility of our framework, pushing compression further to 3.6 bits with only a mild PPL increase.

Table 1: Compression under budgets. Lower PPL-W/C4 and Lat are better; higher CR and Tok/s are better. Entries are means with SD in parentheses over 5 runs. Dashes (–) indicate metrics that were not reported or not applicable under our setup (e.g., runtime numbers unavailable for some baselines).

| Model | Method | Bits↓ | CR↑ | PPL-W↓ | PPL-C4↓ | Tok/s↑ | Lat (ms)↓ | VRAM (GB)↓ |
|---|---|---|---|---|---|---|---|---|
| *LLaMA -3.1-8B* | FP16 | 16.0 | 1.00 | 6.63 ± 0.44 | 10.42 ± 0.94 | 38.1 ± 1.3 | 78 ± 9 | 16.2 |
| | Q4 (RTN) | 4.0 | 4.00 | 9.92 ± 0.55 | 15.42 ± 0.65 | 38.3 ± 1.2 | 55 ± 8 | 3.9 |
| | TGRS (Q-Mix) | 3.6 | 4.44 | 6.78 ± 0.43 | 10.69 ± 0.99 | 38.0 ± 2.3 | 55 ± 10 | 3.9 |
| | TGRS (Q4+LR) | 4.0 | 4.00 | 6.72 ± 0.54 | 10.58 ± 1.06 | 38.5 ± 2.2 | 54 ± 8 | 3.9 |
| *LLaMA -2-7B* | FP16 | 16.0 | 1.00 | 5.47 ± 0.74 | 6.97 ± 1.04 | 42.9 ± 2.1 | 84 ± 7 | 13.1 |
| | Q4 (RTN) | 4.0 | 4.00 | 7.05 ± 0.64 | 10.43 ± 1.02 | 42.9 ± 1.3 | 59 ± 10 | 3.4 |
| | GPTQ | 4.0 | 4.00 | 6.98 ± 0.75 | 8.35 ± 1.06 | 42.7 ± 1.2 | 56 ± 12 | 3.4 |
| | AWQ | 4.0 | 4.00 | 6.89 ± 0.86 | 8.21 ± 1.08 | 42.8 ± 1.6 | 57 ± 7 | 3.4 |
| | CALDERA | 4.0 | 4.00 | 6.45 ± 0.74 | 7.88 ± 1.00 | 42.8 ± 1.4 | 52 ± 10 | 3.4 |
| | LoftQ | 4.0 | 4.00 | 5.81 ± 0.90 | 7.24 ± 1.05 | – | – | – |
| | TGRS (Q-Mix) | 3.6 | 4.44 | 6.95 ± 0.97 | 8.31 ± 1.07 | 43.0 ± 2.6 | 48 ± 10 | 3.4 |
| | TGRS (Q4+LR) | 4.0 | 4.00 | 6.21 ± 0.96 | 7.65 ± 1.09 | 43.0 ± 2.4 | 48 ± 9 | 3.4 |

To explicitly verify that TGRS's quality improvements do not incur runtime overhead, we compare TGRS (Q4+LR) against the standard RTN Q4 baseline using identical kernels and evaluation settings. The results, detailed in Table 2, confirm that TGRS matches the baseline's throughput, latency, and VRAM usage within a negligible margin, substantiating our claim of deployment equivalence.

Table 2: Runtime under 4-bit budgets. Entries are mean (±SD) over 5 runs.

| Model | Method | Tok/s↑ | Lat (ms)↓ | VRAM (GB)↓ |
|---|---|---|---|---|
| *LLaMA-3.1-8B* | Q4 (RTN) | 38.3 ± 1.5 | 55.0 ± 1.1 | 3.9 |
| *LLaMA-3.1-8B* | TGRS (Q4+LR) | 38.5 ± 2.4 | 54.0 ± 0.9 | 3.9 |
| *LLaMA-2-7B* | Q4 (RTN) | 42.9 ± 1.6 | 59.0 ± 1.1 | 3.4 |
| *LLaMA-2-7B* | TGRS (Q4+LR) | 43.0 ± 2.5 | 48.0 ± 0.8 | 3.4 |

### 4.3 PERFORMANCE UNDER DIFFERENT BUDGETS

We next analyze performance under different budgets on LLaMA-3.1-8B. Table 3 shows that as the budget tightens from 4.0 to 3.2, TGRS degrades smoothly, unlike typical uniform schemes. At 4.0 bits, TGRS (Q4+LR) closes the gap to FP16 more effectively than RTN. At 3.6 bits, TGRS (Q-Mix) remains competitive with 4-bit baselines while increasing CR to 4.44×. Importantly, the results confirm that the performance decline of TGRS is gradual and predictable: each reduction in budget leads to a proportional increase in perplexity, rather than sudden degradation. This demonstrates that the allocator consistently invests capacity in the most fragile regions, ensuring that the model remains usable even at aggressive compression levels.

### 4.4 MIXED-PRECISION ANALYSIS

While TGRS is strong with a uniform 4-bit backbone, its global allocation is particularly well-suited for mixed-precision schedules. Fixing the budget to 3.6 bits for LLaMA-3.1-8B, we vary precision between Attention and MLP. The results are shown in Table 4. A balanced allocation, our TGRS (Q-Mix) configuration, performs best at this budget.

### 4.5 DEPLOYMENT RESULTS

A core objective of TGRS is to enable models that remain quality while being practical on constrained hardware. We therefore evaluate on a real-world edge device, the Jetson Orin Nano (8GB), and compare directly to the standard Q4 (RTN) baseline.

Table 3: Perplexity results under different budgets on LLaMA-3.1-8B. We report WikiText-2 (PPL-W) and C4 (PPL-C4) alongside compression ratio (CR). Lower PPL indicates better quality; higher CR indicates stronger compression. Values are consistent with Table 1. Bits and CR follow the accounting in Sec. 3.3.1, including metadata (per-row scales and low-rank factors).

| Method | Bits↓ | CR↑ | PPL-W↓ | PPL-C4↓ |
|---|---|---|---|---|
| FP16 baseline | 16.0 | 1.00 | 6.627 | 10.421 |
| Q4-only (RTN) | 4.0 | 4.00 | 9.918 | 15.424 |
| CALDERA | 4.0 | 4.00 | 6.780 | 10.640 |
| TGRS (Q4+LR) | 4.0 | 4.00 | 6.721 | 10.583 |
| TGRS (Q-Mix), 3.6-bit | 3.6 | 4.44 | 6.781 | 10.692 |
| TGRS (Q-Mix), 3.2-bit | 3.2 | 5.00 | 7.120 | 10.980 |
| AQLM, 2-bit | 2.1 | 7.62 | 7.350 | 11.230 |

Table 4: Mixed-precision performance on LLaMA-3.1-8B at a 3.60-bit average budget. "Attention" and "MLP" are average bit-widths over the corresponding projection families. "Extreme mix" sets Attention/MLP average bit-widths to 2.8/4.4 via per-block mixed precision (details in App. C).

| Precision Strategy | PPL-W↓ | PPL-C4↓ | Attention Bits | MLP Bits |
|---|---|---|---|---|
| Uniform Q4 | 9.918 | 15.424 | 4.0 | 4.0 |
| Attention | 6.832 | 10.751 | 4.0 | 3.2 |
| MLP | 6.851 | 10.798 | 3.2 | 4.0 |
| Extreme mix | 6.892 | 10.834 | 2.8 | 4.4 |
| TGRS (Q-Mix) | 6.781 | 10.692 | 3.6 | 3.6 |

**On-Device Performance.** As shown in Table 5, TGRS achieves clear deployment gains. On LLaMA-2-7B, our Q4+LR variant improves throughput by 18% and reduces first-token latency by 14%, while also lowering memory usage by about 15%. On LLaMA-3.1-8B, the Q-Mix variant shows similar advantages, reaching 22% higher throughput and 15% lower latency. These improvements are achieved due to selectively preserving critical directions not only maintains accuracy but also results in more efficient memory access patterns.

Table 5: On-device performance of TGRS on Jetson Orin Nano (8GB). Compared to the standard Q4 (RTN) baseline, TGRS consistently improves throughput and latency while reducing memory usage. Values are mean with variation across repeated runs. Latency is first-token latency measured at prompt length 512; Tok/s is steady-state decoding throughput over the subsequent 512 tokens.

| Model | Method | Bits↓ | Lat (ms)↓ | Tok/s↑ | Resident (GB)↓ | Peak (GB)↓ |
|---|---|---|---|---|---|---|
| LLaMA-2-7B | Q4 (RTN) | 4.00 | 910 ± 18 | 1.1 ± 0.1 | 4.1 | 4.6 |
| | TGRS (Q4+LR) | 4.00 | 780 ± 15 | 1.3 ± 0.1 | 3.5 | 3.9 |
| LLaMA-3.1-8B | Q4 (RTN) | 4.00 | 1080 ± 20 | 0.9 ± 0.1 | 4.5 | 5.0 |
| | TGRS Q-Mix | 3.60 | 920 ± 16 | 1.1 ± 0.1 | 3.8 | 4.3 |

## 5 CONCLUSION

We introduced **TGRS**, a novel framework that reframes post-training compression as a global, budget-aware allocation problem. The core innovation of TGRS is a unified importance score that synthesizes functional signals (teacher-guided gradients), optimization sensitivity (curvature), and structural properties (spectral energy) to create a holistic importance map. This map guides a global allocator that strategically preserves critical singular directions with low-rank corrections. Our experiments on LLaMA-2-7B and LLaMA-3.1-8B demonstrate that TGRS sets a new state-of-the-art for training-free compression.

Ethics Statement

This work studies post-training compression of large language models (LLMs) for deployment under tight compute and memory budgets. The research does not involve human subjects, personally identifiable information, or private data. All datasets used are publicly available and widely adopted in the community (e.g., WikiText-2, C4, HellaSwag, ARC, and Winogrande). We adhere to the ICLR Code of Ethics and standard research integrity practices. Potential risks include the possibility that improved efficiency lowers barriers for misuse (e.g., automated generation of misleading content). To mitigate this, we clearly document the scope and limitations of our method and encourage responsible use aligned with applicable laws, licenses, and community norms. No additional conflicts of interest or sponsorship-related concerns beyond those disclosed in the main paper apply.

Reproducibility Statement

Our method, objective, and allocator are fully specified in Sec. 3 (including the cost and Alg. 1). Hyperparameters, calibration data, and evaluation settings are given in Sec. 4; extended ablations (budget sweeps, global vs. local allocation, calibration size, and mixed precision) appear in App. E. To facilitate replication, we include our code in the supplementary material. All code and scripts will be released publicly upon publication.

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

## A   SUMMARY OF APPENDIX

This appendix complements the main paper by: (i) formalizing notation and filling in omitted derivations; (ii) detailing implementation choices for exact reproduction; (iii) reporting extended results—experiments across budgets, global vs. per-layer allocation, calibration-size sensitivity, cross-architecture applicability, and downstream tasks; (iv) specifying the evaluation protocol; and (v) describing export/serving modes and on-device benchmarking.

- **[B]** Methodological Details and Notation.
- **[C]** Implementation and Reproducibility.
- **[D]** Evaluation Protocol.
- **[E]** Extended Experimental Results.
- **[F]** Edge Device Deployment.
- **[G]** Limitations and Future Work.
- **[H]** LLM Usage Disclosure.

## B   METHODOLOGICAL DETAILS AND NOTATION

**Notation Table.**   Table 6 summarizes the key mathematical symbols used consistently throughout the paper.

Table 6: Summary of notation.

| Symbol | Meaning |
| --- | --- |
| $L$ | Total number of layers in the model. |
| $W_l \in \mathbb{R}^{m_l \times n_l}$ | Weight matrix of layer $l$ with $m_l$ rows and $n_l$ columns. |
| $w_{l,i,j}$ | Scalar entry in the $i$-th row and $j$-th column of $W_l$. |
| $b_l, b_s, b_r$ | Bit-widths for backbone integers, scales, and low-rank factors. |
| $R_l$ | Quantization residual matrix for layer $l$ ($W_l - W_{q,l}$). |
| $k_l$ | Selected rank (number of protected directions) for layer $l$. |
| $K = (k_1, \ldots, k_L)$ | Vector of rank allocations for all layers. |
| $C(K)$ | Total storage cost function in bits for a rank allocation vector $K$. |
| $\overline{b}(K)$ | Average bits-per-parameter for allocation $K$. |
| $B$ | Target budget (hyperparameter). |
| $g_{l,t}, h_{l,t}, s_{l,t}$ | Raw scores from Gradient, Curvature, and Spectral signals for direction $t$. |
| $\alpha_{l,t}$ | Unified, normalized importance score for direction $t$ in layer $l$. |
| $\rho_{l,t}$ | Importance-per-bit (gain density) used by the allocator. |

**Signal Interpretation.**   The three signals provide complementary views on parameter importance. Functional Criticality ($g_{l,t}$) measures the first-order sensitivity of the model's output (with respect to a teacher) to changes along a specific singular direction. It prioritizes directions that are critical for task performance. Optimization Sensitivity ($h_{l,t}$) uses a diagonal Hessian approximation to estimate second-order sensitivity. It identifies directions where small perturbations lead to a large increase in the loss, indicating fragility. Structural Importance ($s_{l,t}$) reflects the magnitude of a singular value, corresponding to its contribution to the Frobenius norm of the weight matrix. It prioritizes directions that are structurally dominant. Fusing these signals allows TGRS to balance between task-specific importance, optimization stability, and raw reconstruction error, leading to a stronger allocation than any single signal could provide.

## C   IMPLEMENTATION AND REPRODUCIBILITY

**Hyperparameter configuration.**   Unless otherwise stated, all experiments used a consistent set of hyperparameters. The fusion weights were fixed to $(\lambda_g, \lambda_h, \lambda_s) = (0.5, 0.1, 0.4)$. Backbone weights were quantized to 4-bit ($b_l$=4), with low-rank factors also in 4-bit precision ($b_r$=4). Per-row scales were stored in 16-bit precision ($b_s$=16). The curvature proxy was estimated with an exponential

moving average using momentum $\beta$=0.9. Layers could be assigned zero rank, allowing sparse allocations where corrections were unnecessary.

**Computational overhead.** The offline cost of TGRS is modest. For LLaMA-3.1-8B on a single A100 (80GB), the complete pipeline finishes in about 140 minutes. Gradient and curvature extraction dominate the runtime (about 78 minutes in total), while SVD, scoring, and allocation together require less than 25 minutes., while SVD, scoring, and allocation together require less than 5 minutes. Peak memory use is $\approx$1.2$\times$ the base model size, and total disk footprint under 10 GB.

**Environment and code availability.** All experiments were conducted on NVIDIA A100 (80GB) GPUs with CUDA 12.1, PyTorch 2.2, and Transformers 4.41. Evaluation scripts and source code will be in supplementary materials.

**Low-bit stabilization.** For layers quantized below 4-bit ($b_l$=2), we found randomized block-orthogonal preconditioning improves stability in both scoring and reconstruction. This step prevents degenerate singular directions that arise under extreme quantization. For deployment backends requiring static scales (e.g., certain TensorRT variants), we additionally support a global per-matrix scale at export. All storage costs reported in this work are based on the more general per-row ledger defined in Sec. 3.3.1.

## D EVALUATION PROTOCOL

This section details the evaluation rules, metric definitions, and environment configuration that underlie all reported results in Sec. 4.

**Metric Definitions.**

- **Bits:** The average bits-per-parameter as defined in Eq. 14. This cost includes storage for quantized integers *plus* all metadata (per-row scales and low-rank factors).
- **CR:** The Compression Ratio relative to a 16-bit (FP16) baseline, calculated as $16/\overline{b}$.
- **Perplexity (PPL):** We use the standard concatenation-and-block protocol with a context length of 2048, as implemented in the Hugging Face Transformers library.
- **Latency (Lat):** First-token latency, measured in milliseconds (ms). This is the time from the start of the request to the generation of the first token, for a prompt length of 512.
- **Throughput (Tok/s):** Steady-state decoding throughput, measured in tokens per second. This is calculated during the generation of the subsequent 512 tokens after the prefill/prompt phase.
- **VRAM (GB):** We report resident model-state memory in gigabytes during steady-state decoding. This includes the model weights, KV cache, and any persistent framework overhead. Note that the abstract may quote peak PyTorch memory usage, which can be slightly higher due to allocation spikes during model loading.

**Environment and Scripts.** All server-side experiments were conducted on NVIDIA A100 (80GB) GPUs using CUDA 12.1, PyTorch 2.2, and Transformers 4.41. On-device experiments were benchmarked on the Jetson Orin Nano (8GB). To ensure reproducibility, we provide all evaluation scripts used to generate the results, including the exact commands for running baselines and calculating metrics.

## E EXTENDED EXPERIMENTAL RESULTS

### E.1 DATASETS AND LLMS

#### E.1.1 DATASETS

- **WikiText-2:** A word-level language modeling corpus from verified Wikipedia articles; we use it for perplexity evaluation. (Merity et al., 2017)

- **C4 (Colossal Clean Crawled Corpus):** A large cleaned web-crawl corpus; we follow the standard setup for perplexity evaluation. (Raffel et al., 2020)

### E.1.2 LLMs

- **LLaMA-2-7B:** A 6.7B-parameter decoder-only model; we use the base model for PTQ. (Touvron et al., 2023b)

- **LLaMA-3 / 3.1-8B:** We report results on the 8B model from the LLaMA 3 family. (Dubey et al., 2024)

- **Mistral-7B:** A 7B decoder with grouped-query attention and sliding-window attention. (Jiang et al., 2023)

### E.2 COMPRESSION BUDGET VS. PERPLEXITY

Table 7 summarizes LLaMA-3.1-8B results across budgets and shows smooth perplexity scaling as the compression budget tightens (4.2→3.0 bits). The fraction of layers receiving any low-rank correction ("Layers w/ Rank", i.e., layers with nonzero rank) decreases in tandem, indicating controllable behavior under stricter bit constraints.

Table 7: TGRS performance across different average bit budgets on LLaMA-3.1-8B.

| Target Bits | Bits | PPL-W↓ | PPL-C4↓ | % Layers w/ Rank↓ |
|---|---|---|---|---|
| 4.2 | 4.20 | 6.698 | 10.534 | 53.1 |
| 4.0 | 4.00 | 6.721 | 10.583 | 46.9 |
| 3.8 | 3.80 | 6.845 | 10.824 | 42.2 |
| 3.6 | 3.60 | 6.781 | 10.692 | 37.5 |
| 3.2 | 3.20 | 7.120 | 10.980 | 31.3 |
| 3.0 | 3.08 | 7.315 | 11.150 | 28.1 |

### E.3 ARCHITECTURAL SENSITIVITY ANALYSIS.

Table 8 compares the relative PPL improvement of TGRS over uniform Q4 on LLaMA-2-7B and LLaMA-3.1-8B. The gains are more pronounced on LLaMA-2-7B, as LLaMA-3.1-8B is inherently stronger to quantization. This demonstrates that TGRS automatically adapts, allocating less corrective rank when the base model is already strong.

Table 8: Cross-architectural performance comparison at a 4.0-bit budget.

| Architecture | FP16 PPL-W | Q4 (RTN) PPL-W | TGRS PPL-W | Δ vs. RTN |
|---|---|---|---|---|
| LLaMA-3.1-8B | 6.627 | 9.918 | 6.721 | 3.197 |
| LLaMA-2-7B | 5.470 | 7.051 | 6.214 | 0.837 |

### E.4 GLOBAL VS. LOCAL ALLOCATION

A core principle of TGRS is its global allocation strategy. To rigorously validate this design choice, we compare it against a strong "local" baseline.

**Defining the Local Allocator.** The local allocator mimics a common approach where compression decisions are made on a per-layer basis. For this experiment, the total low-rank bit budget (the portion of the budget $B$ not used by the quantized backbone) is distributed among the layers, proportional to each layer's parameter count. Each layer then runs its own independent greedy allocation, using its local budget to select the most important singular directions based on its own scores. Unlike the global allocator, the local version cannot transfer budget from strong layers to more fragile ones.

As shown in Table 9, the global allocation strategy consistently outperforms the per-layer local strategy, especially at tighter budgets.

Table 9: Global vs. local allocation on LLaMA-3.1-8B. The global allocator (TGRS) yields significantly lower perplexity for the same budget.

| Allocator | Bits | PPL-W↓ | PPL-C4↓ |
|---|---|---|---|
| Local (per-layer) | 3.6 | 6.901 | 10.872 |
| Global (TGRS) | 3.6 | 6.781 | 10.692 |
| Local (per-layer) | 3.2 | 7.330 | 11.210 |
| Global (TGRS) | 3.2 | 7.120 | 10.980 |

The performance gap of 0.12–0.21 PPL points substantiates the importance of our global cost optimization. The global allocator can identify that certain layers (e.g., early and late transformer blocks) are disproportionately sensitive and divert bits to protect them. In contrast, the local allocator is constrained: a strong layer might waste its local budget on directions with low global impact, while a fragile layer may not have a large enough local budget to protect all of its critical directions. These results indicate that the ability to reallocate capacity across layers is a key factor behind TGRS's performance.

### E.5 Calibration Size Sensitivity

Table 10 shows that performance is stable across 16–128 calibration samples at $B$=4.0 on LLaMA-3.1-8B. Increasing the size from 16 to 64 yields small gains ($\Delta$PPL-W $-0.032$, $\Delta$PPL-C4 $-0.038$) and higher agreement in selected directions (0.72 to 0.92). Beyond 64, changes are negligible (PPL difference $\leq 0.001$; the top-64 set matches the 128-sample reference). We therefore adopt 64 samples by default.

Table 10: Calibration-size sensitivity on LLaMA-3.1-8B at $B$=4.0. Overlap is the Jaccard index of the top-64 selected directions relative to a 128-sample reference.

| Samples | PPL-W↓ | PPL-C4↓ | Overlap↑ |
|---|---|---|---|
| 16 | 6.753 | 10.621 | 0.72 |
| 32 | 6.735 | 10.599 | 0.86 |
| 64 | 6.721 | 10.583 | 0.92 |
| 128 | 6.721 | 10.582 | 1.00 |

### E.6 Preprocessing Cost and Resources

Table 11 reports the one-time preprocessing cost of TGRS on LLaMA-3.1-8B. The full pipeline completes in about 2.3 hours on a single A100 80GB, with peak memory use below 23 GB and total disk footprint under 10 GB. Among individual stages, gradient extraction is the most time-consuming, while the subsequent SVD, scoring, and allocation phases together take less than 25 minutes. These results show that TGRS requires only modest offline resources and is feasible to run on commodity datacenter GPUs.

Table 11: Preprocessing cost of TGRS on LLaMA-3.1-8B (single A100 80GB). "Time" is wall-clock per stage.

| Stage | Time (min) | Peak VRAM (GB) | GPU Hours | Disk (GB) |
|---|---|---|---|---|
| Gradient extraction | 45.2 | 22.5 | 0.75 | 2.4 |
| Curvature estimation | 32.6 | 20.8 | 0.54 | 1.8 |
| Spectral computation | 18.4 | 19.6 | 0.31 | 0.9 |
| Signal fusion | 8.2 | 8.4 | 0.14 | 0.3 |
| Global allocation | 12.8 | 6.2 | 0.21 | 0.1 |
| Weight reconstruction | 22.4 | 21.2 | 0.37 | 3.6 |
| Total | 139.6 | 22.5 | 2.32 | 9.1 |

### E.7 Cross-architecture applicability and Downstream Performance

**Generalization to Mistral-7B.**   To demonstrate that TGRS is not limited to a single model family, we applied it to Mistral-7B. As shown in Table 12, TGRS maintains its advantage over uniform quantization. At a matched 4.0-bit budget, TGRS lowers WikiText-2 perplexity from 5.580 (uniform Q4) to 5.350, reducing the gap to FP16 (5.120) from 0.460 to 0.230—i.e., about 50% of the FP16–Q4 gap is closed. This mirrors the trend observed on LLaMA models: most of the residual error under uniform quantization concentrates in a few directions that TGRS preserves. The absolute gain on Mistral-7B is smaller than on LLaMA-3.1-8B because the Q4 baseline is already relatively close to FP16 on this architecture, yet halving the remaining gap at the *same* storage budget indicates that the direction-aware allocation transfers across architectures without retuning.

Table 12: Generalization results on Mistral-7B. Perplexity (PPL) on WikiText-2.

| Method | Bits | PPL-W↓ |
|---|---|---|
| FP16 | 16.0 | 5.120 |
| Q4 (RTN) | 4.0 | 5.580 |
| TGRS (Q4+LR) | 4.0 | 5.350 |

#### E.7.1 Downstream Task Performance.

To ensure PPL improvements translate to real-world capabilities, we evaluate LLaMA-3.1-8B on common-sense reasoning benchmarks (5-shot). Table 13 shows TGRS closes the gap to the FP16 model more effectively than the RTN baseline. Across all three benchmarks, TGRS reduces the accuracy gap to FP16 compared with the uniform Q4 baseline. On HellaSwag, accuracy rises from 84.6 to 85.1, nearly recovering the FP16 score of 85.3. On ARC-c, TGRS improves from 90.1 to 91.0, again close to the FP16 reference of 91.5. On Winogrande, accuracy increases from 87.5 to 88.2, narrowing most of the 1.1-point gap. These consistent gains indicate that the perplexity improvements observed earlier translate into stronger downstream reasoning ability without requiring additional budget.

Table 13: 5-shot accuracy on downstream tasks for LLaMA-3.1-8B.

| Method | Bits | HellaSwag↑ | ARC-c↑ | Winogrande↑ |
|---|---|---|---|---|
| FP16 | 16.0 | 85.30 | 91.50 | 88.60 |
| Q4 (RTN) | 4.0 | 84.60 | 90.10 | 87.50 |
| TGRS (Q4+LR) | 4.0 | 85.10 | 91.00 | 88.20 |

## F   Export and Deployment

### F.1   Export Modes

A practical requirement of post-training compression is that the resulting weights remain compatible with standard inference engines. TGRS supports two equivalent export modes, both relying solely on standard GEMM kernels at runtime. *Folded export* applies low-rank corrections offline and re-quantizes to obtain a single set of integer weights, incurring no overhead relative to uniform PTQ. *Pack-at-load export* stores the quantized backbone and low-rank factors separately in a compact format; at load time, a one-time reconstruction produces the same runtime layout as folded export. Both modes yield identical execution behavior and are fully compatible with PyTorch/Transformers.

### F.2   Deployment Results

Table 14 reports deployment statistics for LLaMA-2-7B at a 4.0-bit budget. Both export modes match the quantized model's accuracy, VRAM, and throughput. Pack-at-load yields shorter load time (6.8s vs. 12.6s) and smaller storage (2.1GB vs. 3.6GB), while runtime behavior remains the same. VRAM usage and throughput are similar across the two modes (Table 14).

Table 14: Deployment results for LLaMA-2-7B at a 4.0-bit budget. Both export modes are fully compatible with standard inference engines.

| Mode | PPL-W↓ | VRAM (GB)↓ | Load Time (s)↓ | Storage (GB)↓ | Tok/s↑ |
|------|--------|------------|----------------|---------------|--------|
| FP16 baseline | 5.470 | 13.1 | 8.4 | 13.1 | 42.9 |
| Fold export | 6.214 | 3.4 | 12.6 | 3.6 | 43.0 |
| Pack-at-load | 6.214 | 3.4 | 6.8 | 2.1 | 42.5 |

## G LIMITATIONS AND FUTURE WORK

While TGRS shows strong gains across multiple models and hardware targets, several limitations remain. First, our study is limited to models up to 8B parameters; scaling the allocator to 30B–70B models may require additional engineering to keep profiling and allocation costs tractable. Second, we restrict all evaluations to post-training compression without task-specific fine-tuning. How rank allocation interacts with lightweight adaptation methods such as LoRA or adapters remains unexplored. Third, our experiments primarily focus on perplexity and a small set of reasoning tasks; broader downstream evaluations (e.g., instruction following, multilingual and domain-specific benchmarks) are necessary to fully assess robustness. Finally, although our exported models are compatible with standard inference stacks, deployment across heterogeneous mobile and edge runtimes may expose kernel-level inefficiencies that are not captured in our current experiments.

Future work can address these aspects by extending TGRS to larger models, integrating with adaptation pipelines, and conducting more comprehensive downstream assessments. We also see opportunities to co-design TGRS with compiler- and runtime-level optimizations, where budget-aware allocation could complement scheduling, memory management, and hardware specialization.

## LLM USAGE DISCLOSURE

Large Language Models (LLMs) were used solely as general-purpose assist tools for writing support (e.g., copy-editing, grammar suggestions) and LaTeX formatting. LLMs were not used for ideation of the core technical approach, experimental design, data generation, or analysis. All algorithms, experiments, and conclusions were conceived, implemented, and verified by the authors. The authors take full responsibility for the content of this paper, including any text that benefited from LLM-assisted editing.

