# OpenReview forum: "TGRS: Teacher-Guided Rank-Sensitive Quantization for Large Language Models"
_ICLR.cc/2026/Conference — ICLR 2026 Conference Withdrawn Submission_

### Official Review · Reviewer_f6C2 · 2025-10-28

**Soundness:** 2
**Presentation:** 2
**Contribution:** 2
**Rating:** 2
**Confidence:** 4

**Summary:**

This paper proposes Teacher-Guided Rank-Sensitive Quantization (TGRS), a post-training compression framework for large language models. It formulates quantization and low-rank correction as a global budget-aware allocation problem, where parameter importance is estimated from three complementary signals — teacher-guided gradients, curvature (optimization sensitivity), and spectral energy (structural importance). These are fused into a unified score that drives a global allocator to assign precision/rank selectively across layers. The resulting model combines a uniformly quantized backbone with low-rank residual corrections, achieving competitive ppl and improved efficiency on edge devices.

**Strengths:**

- The paper presents various scoring and allocation strategies to utilize the compression budget more effectively from different perspectives.
- The proposed method shows efficiency and deployability through on-device experiments.

**Weaknesses:**

- The idea of initializing low-rank components using quantization error is very similar to LoftQ, and when directly comparing ppl with LoftQ, TGRS is worse, which weakens the claimed improvement.
- The fusion of scoring signals uses fixed coefficients (0.5/0.1/0.4). There is no analysis or sweep to justify that these hyper-parameters are optimal, nor comparisons against alternative settings
- Experimental evaluation is limited: the tested models and tasks are too few, and almost all results are on PPL. Additional benchmarks like Commonsense QA or GSM8K (open-ended generation) would help demonstrate robustness of proposed method.
- The overall pipeline is relatively complex compared to previous baselines, involving multiple stages and teacher model usage, which may limit its practicality for edge deployment.

**Questions:**

In Table 5, TGRS shows faster speed than RTN even with the same average bit width. It would be helpful if the authors could explain how such acceleration is achieved while maintaining the same bit budget.

---

### Official Review · Reviewer_csfL · 2025-10-30

**Soundness:** 3
**Presentation:** 3
**Contribution:** 2
**Rating:** 4
**Confidence:** 4

**Summary:**

This paper introduces TGRS, a training-free compression framework for LLMs. Its core contribution is a "unified sensitivity score" that synthesizes three signals(teacher-guided gradients, optimization-based curvature, and structural spectral energy) to holistically identify critical model components . This score guides a global, budget-aware allocator that adds selective low-rank corrections to a uniformly quantized backbone. Experiments show TGRS achieves SOTA perplexity at fixed budgets and good performance on edge devices.

**Strengths:**

1. This paper tackles a critical problem: the deployment of LLMs on edge devices.
2. The method is training-free, which allows for good practicality.
3. The paper is well-written. The 3-stage framework (Scoring, Allocation, Reconstruction) provides a clear logical flow for its methodology.

**Weaknesses:**

1. The paper lacks a more detailed literature review on global budget allocation for LLM compression. Most relevantly, BitStack[1] also allows layer-wise memory budget allocation using iterative SVD and matches quantization quality. AnyPrecision LLM[2] and Matryoshka Quantization[3] allow different bit-widths for different layers in quantization. I believe these papers should at least be cited and discussed.
2. The evaluations could benefit from real-world downstream tasks, instead of only evaluating on perplexity. This would add value to the practicality of the method.
3. The scalability is unclear, The profiling cost is non-trivial and could be a bottleneck at a scale of 70B or larger.

[1] BitStack: Any-Size Compression of Large Language Models in Variable Memory Environments

[2] Any-Precision LLM: Low-Cost Deployment of Multiple, Different-Sized LLMs

[3] Matryoshka Quantization

**Questions:**

See weaknesses.

---

### Official Review · Reviewer_oqRe · 2025-10-31

**Soundness:** 2
**Presentation:** 2
**Contribution:** 2
**Rating:** 4
**Confidence:** 4

**Summary:**

The paper introduces TGRS (Teacher-Guided Rank-Sensitive quantization), a post-training compression framework that combines a uniformly quantized backbone with selectively added low-rank corrections chosen under a global bit-budget. The key idea is to score singular directions using three signals computed post-hoc on a small calibration set: (i) teacher-guided gradients from a KD loss, (ii) a curvature proxy (EMA of squared gradients) and (iii) spectral energy; these are min-max normalized and linearly combined into a unified importance score. A greedy knapsack-style allocator then spends the available budget on directions with the highest importance-per-bit, and the model is reconstructed by adding the top-(k) residual SVD components to the quantized weights. On LLaMA-3.1-8B, the paper claims 3.6 average bits with PPL(WikiText-2) 6.78 vs FP16 6.63 and total memory ≈3.9 GB, and reports comparable runtime to Q4 RTN along with on-device results on Jetson Orin Nano.

**Strengths:**

* Clear global budget accounting that includes metadata costs and defines average bits precisely; this is often glossed over and is a real contribution to fair comparison. ()
* Unified direction-level score combining functional (teacher gradients), optimization (curvature proxy), and structural (spectral energy) signals; simple and practical to compute post-training.
* Greedy importance-per-bit allocator is straightforward and reproducible (Algorithm 1).
* Empirical wins at 4-bit budgets on LLaMA-2-7B and a 3.6-bit variant on LLaMA-3.1-8B, with small PPL deltas vs FP16 and matched throughput to RTN Q4 in their setup.
* Edge-device evaluation on Jetson Orin Nano with latency and VRAM numbers is welcome and atypically practical.

**Weaknesses:**

* Baseline coverage is uneven. Table 1 for LLaMA-3.1-8B compares mainly to RTN, whereas 7B includes GPTQ/AWQ/CALDERA/LoftQ. The absence of these strong baselines at 8B makes it difficult to assess state-of-the-art positioning. Provide 8B results for the same set.
* Limited task breadth. The core metric is perplexity on WikiText-2 and C4; downstream accuracy (HellaSwag/ARC/Winogrande etc.) is not reported in the main tables. Per the ethics section the datasets are standard, but main-paper evidence focuses on PPL; add zero-shot task results to validate that direction-level protection generalizes beyond PPL.
* Computational overhead and scalability not analyzed. Two SVDs per layer plus gradient and curvature accumulation could be costly at 8B and beyond. No wall-clock or memory profiling for the compression pipeline is provided; no discussion of randomized/blocked SVD to reduce cost.
* Runtime claims need ablation. The paper claims matched throughput/VRAM relative to RTN despite extra low-rank matmuls. Without kernel-level profiling or explanation of memory-access benefits, the reported on-device speedups read optimistic. Include profiles or ablate the LR path to show where time is saved.
* Hyperparameters and sensitivity. The fixed mixing weights ((\lambda_g,\lambda_h,\lambda_s)=(0.5,0.1,0.4)) are not justified; sensitivity to these weights and to calibration size/domain is only briefly referenced. Add ablations in the main paper.
* Mixed-precision vs hybrids. While a 3.6-bit mix is reported, comparisons against recent sub-4-bit PTQ and hybrid methods are incomplete in the 8B setting; ensure apples-to-apples under the same budget accounting.

**Questions:**

1. Pipeline cost: What is the wall-clock time and peak memory to run TGRS end-to-end on LLaMA-3.1-8B? Please report per-stage times (scoring SVD, gradient/EMA accumulation, allocator, residual SVD) and whether randomized SVD is used.
2. Allocator optimality: The greedy density allocator is intuitive, but how far is it from the true optimum? Any comparison to a DP/ILP solver on smaller models to bound sub-optimality?
3. Runtime equivalence: Please provide kernel-level profiling that explains why adding rank-(k) residuals does not hurt throughput relative to RTN Q4, and how the memory-access pattern changes.
4. Calibration sensitivity: How does performance vary with calibration size and domain mismatch (e.g., no instruction-tuned data)? Provide a domain-shift study.
5. Downstream tasks: Include main-paper results on HellaSwag/ARC/Winogrande to complement PPL and validate task-level robustness at fixed bits.
6. Hyperparameters: Provide an ablation over ((\lambda_g,\lambda_h,\lambda_s)) and justify the chosen weights; also examine the effect of replacing one signal at a time.
7. Budget accounting across methods: Confirm that all baselines are re-accounted under your metadata-inclusive definition; otherwise the CR vs bits comparison may be biased.

---

### Official Review · Reviewer_eWiD · 2025-11-08

**Soundness:** 1
**Presentation:** 2
**Contribution:** 1
**Rating:** 2
**Confidence:** 4

**Summary:**

This paper introduces Teacher-Guided Rank-Sensitivity (TGRS), a post-training compression framework for Large Language Models (LLMs). The method recasts compression as a global, budget-aware allocation problem. Its core contribution is a "Unified Importance Score," which synthesizes functional (teacher-guided gradients), optimization (curvature proxy), and structural (spectral energy) signals to quantify the sensitivity of different parameter directions. This score guides a global allocator to apply selective low-rank (LR) corrections to a uniformly quantized backbone. The authors claim this approach preserves model fidelity under aggressive bit-budgets (e.g., 3.6-bit) and can be deployed on standard hardware without custom kernels.

**Strengths:**

- Significance: The paper tackles the critical and high-impact problem of deploying LLMs on resource-constrained edge devices.

- Originality: The formulation of post-training compression as a global knapsack problem (Eq 15) guided by a multi-faceted importance score is a creative and principled combination of existing concepts.

- Clarity: The proposed 3-stage framework (scoring, allocation, reconstruction) is articulated adequately in Figure 1 and Algorithm 1.

**Weaknesses:**

- The paper's core contribution, the "Unified Importance Score" (Eq 11), is a weighted sum of three signals. However, the paper provides no quantitative analysis, ablation study, or justification for the specific weights $(\lambda_g=0.5, \lambda_h=0.1, \lambda_s=0.4)$. Without this, the method's key component appears to be based on heuristic tuning rather than a derivable principle, which weakens its generality.

- Scalability and Practicality: The method's practicality for state-of-the-art LLMs is questionable. The profiling stage, required to compute gradients and curvature, is computationally intensive (e.g., 78 minutes on an A100 for an 8B model, as shown in Table 11). This cost will likely become prohibitive for models in the 70B+ range. Furthermore, the experiments are confined to 8B models. The paper itself concedes in Appendix G that scaling to 30B-70B models is a non-trivial challenge, leaving the applicability of its approach to current SOTA models unproven.

- Experimental Comparisons and Performance: When compared to CALDERA, a more direct (hybrid Quant + LR) competitor, the performance gain is marginal (e.g., 6.721 vs. 6.780 PPL-W, a 0.059 difference, in Table 3). This minimal improvement may not justify the added complexity of TGRS (requiring three signals vs. CALDERA's one). Furthermore, the results do not consistently show superiority. In Table 1, the LoftQ baseline achieves a 5.81 PPL on LLaMA-2-7B, which is significantly better than the proposed TGRS (Q4+LR) score of 6.216.

**Questions:**

- Could the authors provide an ablation study or sensitivity analysis for the hyperparameters in Equation 11? How were the $(\lambda_g=0.5, \lambda_h=0.1, \lambda_s=0.4)$ weights determined, and how much does performance vary with different values?

- Given the high offline cost (78 minutes for an 8B model), what is the authors' concrete proposal for scaling this method to 70B or 100B+ models? Are there cheaper proxies for the gradient and curvature signals that could be used?

---

### Note · Authors · 2025-11-13

I have read and agree with the venue's withdrawal policy on behalf of myself and my co-authors.